# Milk Intake and Stroke Mortality in the Japan Collaborative Cohort Study—A Bayesian Survival Analysis

**DOI:** 10.3390/nu12092743

**Published:** 2020-09-09

**Authors:** Chaochen Wang, Hiroshi Yatsuya, Yingsong Lin, Tae Sasakabe, Sayo Kawai, Shogo Kikuchi, Hiroyasu Iso, Akiko Tamakoshi

**Affiliations:** 1Department of Public Health, Aichi Medical University School of Medicine, 480-1195 Aichi, Japan; linys@aichi-med-u.ac.jp (Y.L.); tsasa@aichi-med-u.ac.jp (T.S.); ksayo@aichi-med-u.ac.jp (S.K.); kikuchis@aichi-med-u.ac.jp (S.K.); 2Department of Public Health, Fujita Health University School of Medicine, Toyoake 470-1192, Japan; yatsuya@fujita-hu.ac.jp; 3Public Health, Department of Social Medicine, Osaka University Graduate School of Medicine, Osaka 565-0871, Japan; iso@pbhel.med.osaka-u.ac.jp; 4Department of Public Health, Hokkaido University, Faculty of Medicine, Sapporo 002-8501, Japan; tamaa@med.hokudai.ac.jp

**Keywords:** milk intake, mortality, stroke, Bayesian survival anlysis, time-to-event data, JACC study

## Abstract

The aim of this study was to further examine the relationship between milk intake and stroke mortality among the Japanese population. We used data from the Japan Collaborative Cohort (JACC) Study (total number of participants = 110,585, age range: 40–79) to estimate the posterior acceleration factors (AF) as well as the hazard ratios (HR) comparing individuals with different milk intake frequencies against those who never consumed milk at the study baseline. These estimations were computed through a series of Bayesian survival models that employed a Markov Chain Monte Carlo simulation process. In total, 100,000 posterior samples were generated separately through four independent chains after model convergency was confirmed. Posterior probabilites that daily milk consumers had lower hazard or delayed mortality from strokes compared to non-consumers was 99.0% and 78.0% for men and women, respectively. Accordingly, the estimated posterior means of AF and HR for daily milk consumers were 0.88 (95% Credible Interval, CrI: 0.81, 0.96) and 0.80 (95% CrI: 0.69, 0.93) for men and 0.97 (95% CrI: 0.88, 1.10) and 0.95 (95% CrI: 0.80, 1.17) for women. In conclusion, data from the JACC study provided strong evidence that daily milk intake among Japanese men was associated with delayed and lower risk of mortality from stroke especially cerebral infarction.

## 1. Introduction

Eastern Asian populations were reported to have a higher burden from both mortality and morbidity from stroke than populations in European or American regions [1]. Dairy foods, especially milk, have been suggested to relate with decreased stroke risk by nearly 7% for each 200 g increment of daily consumption [2]. Although two daily servings of milk or dairy products is recommended in Japan [3], the actual per capita intake (≈63 g/day) of these food groups is much lower and less frequent than that in Western countries [4]. Given that previous reports have also indicated no significant [5,6,7] or even positive associations [8], it would be of interest to provide information on the association within a context where most individuals have much lower range of milk intake compared to most of the previous studies. Whether people with such a low level intake of milk can still benefit against stroke would requires elucidation.

Moreover, a more intuitive or straightforward interpretation would be available if we were able to show the probabilities and the certainty that the existing data can provide evidence about whether drinking milk can delay or lower the hazard of dying from stroke. This would require the transformation of the research question into “Do individuals with milk intake habit have lower hazard of dying from a stroke event?” Another way of asking the same research question would be “Can an event of dying from stroke be postponed by forming a milk intake habit?” The later question directly compares the length of times before events or the speed from the same departure point (entry of the study) to a fatal stroke event. Accelerated failure time (AFT) models under the Bayesian framework are convenient tools that would help to avoid worrying about the assumptions of repeatability of a large scale prospective study, and can calculate the exact proportion of posterior estimates (hazard ratios, HR) that are lower than 1. AFT model is helpful in estimating acceleration factors (AF) which can be interpreted as the velocity for developing an event of interest. This velocity parameter shows how faster/slower individuals in one exposure group might have an event compared to others in different exposure groups [9,10].

Our aim was to provide a more straightforward answer to the primary research question that, whether someone answering that he/she drank milk at the baseline of study, had lower hazard of dying from stroke compared with his/her counterparts who said they never consumed milk. If the answer to the primary objective was yes, then the probabilities that individuals with different frequencies of milk intake may have lower risk compared with those who never drank milk were calculated through a Markov Chain Monte Carlo (MCMC) simulation process. A Bayesian survival analysis method was applied on an existing database and through which we provided estimates about whether drinking milk could delay a stroke mortality event from happening after controlling for the other potential confounders.

## 2. Materials and Methods

### 2.1. The Database

We used data from the Japan Collaborative Cohort (JACC) study, which was sponsored by the Ministry of Education, Sports, Science, and Technology of Japan. Sampling methods and details about the JACC study have been described extensively in the literature [11,12,13]. Participants in the JACC study completed self-administered questionnaires about their lifestyles, food intake (food frequency questionnaire, FFQ), and medical histories of cardiovascular disease or cancer. In the final follow-up of the JACC study, data from a total of 110,585 individuals (46,395 men and 64,190 women) who, aged between 40–79 at the baseline, were successfully retained for the current analysis. We further excluded samples if they met one of the following criteria: (1) with any disease history of stroke, cancer, myocardial infarction, ischemic heart disease, or other types heart disease (*n* = 6655, 2931 men and 3724 women); (2) did not answer the question regarding their milk consumption in the baseline FFQ survey (*n* = 9545, 3593 men and 5952 women). Finally, 94,385 (39,386 men and 54,999 women) were left in the database. The study design and informed consent procedure were approved by the Ethics Review Committee of Hokkaido University School of Medicine.

### 2.2. Exposure and the Outcome of Interest

Frequency of milk intake during the preceding year of the baseline was assessed by FFQ as “never”, “1–2 times/month”, “1–2 times/week”, “3–4 times/week”, and “Almost daily”. The exact amount of milk consumption was difficult to assess here. However, good reproducibility and validity were confirmed previously (Spearman rank correlation coefficient between milk intake frequency and weighed dietary record for 12 days was 0.65) [14]. From the same validation study [14] on the FFQ, daily consumers of milk were found to have a median intake of 146 g. At the baseline of the JACC study (between 1988 and 1990), most “milk and dairy product” consumption (92.1%) was in the form of whole milk [15].

In the study area, investigators conducted a systematic review of death certificates till the end of 2009. Date and cause of death were confirmed with the permission of the Director-General of the Prime Minister’s Office. The follow-up period was defined as from the time the baseline survey was completed, which was between 1988–1990, until the end of 2009 (administrative censor), the date when they moved out of the study area, or the date of death from stroke was recorded—whichever occurred first. Other causes of death were treated as censored and assumed not informative. The causes of death were coded by the 10th Revision of the International Statistical Classification of Diseases and Related Health Problems (ICD-10); therefore, stroke was defined as I60–I69. We further classified these deaths into hemorrhagic stroke (I60, I61 and I62) or cerebral infarction (I63) when subtypes of stroke in their death certificates were available.

### 2.3. Statistical Approach

Analyses were stratified by sex as difference in milk consumption levels and mortality rate were suggested previously [16,17]. We calculated sex-specific means (standard deviation, SD) and proportion of selected baseline characteristics according to the frequency of milk intake. Age-adjusted stroke mortality rate were expressed as per 1000 person-year, predicted through poisson regression models.

Full parametric proportional hazard models under Bayesian framework with Weibull distribution were fitted using Just Another Gibbs Sampler (JAGS) program [18], version 4.3.0 in R, version 4.0.1 [19]. JAGS program is similar to the OpenBUGS [20] project that uses a Gibbs sampling engine for MCMC simulation. In the current analysis, we specified non-informative prior distributions for each of the parameters in our models ($\beta_{n}∼N\left(0,1000\right)$, and $\kappa_{\mathrm{shape}}∼\Gamma \left(0.001,0.001\right)$). The Brooks–Gelman–Rubin diagnostic [21] was used to refine the approximate point of convergence, the point when the ratio of the chains is stable around 1 and the within and between chain variability start to reach stability, was visually checked. The auto-correlation tool further identified if convergence has been achieved or if a high degree of auto-correlation exists in the sample. Then, the number of iterations discarded as “burn-in” was chosen. All models had a posterior sample size of 100,000 from four separated chains with a “burn-in” of 2500 iterations. Posterior means (SD) and 95% Credible Intervals (CrI) of the estimated hazard ratios (HRs), as well as acceleration factors (AFs), were presented for each category of milk intake frequency taking the “never” category as the reference. Posterior probabilities that the estimated hazard of dying from stroke for the milk intake for frequency that higher or equal to “1–2 times/month” were smaller compared with those who chose “never” for their milk intake frequency, and were calculated as Pr(HR<1).

The parametric forms of the models fitted in the Bayesian survival analyses included three models: (1) the crude model, (2) the age-centered adjusted model, (3) and a model further adjusted for potential confounders which includes: age (centered, continuous), smoking habit (never, current, former), alcohol intake (never or past, <4 times/week, Daily), body mass index (<18.5, ≥18.5 and <25, ≥25 and <30, ≥30 kg/m^2^), history of hypertension, diabetes, kidney/liver diseases (yes/no), exercise (more than 1 h/week, yes/no), sleep duration (<7, ≥7 and <8, ≥8 and <9, ≥9, hours), quartiles of total energy intake, coffee intake (never, <3–4 times/week, almost daily), and education level (attended school till age 18, yes/no).

## 3. Results

The total follow-up was 1,555,073 person-year (median = 19.3 years), during which 2675 deaths from stroke were confirmed (1352 men and 1323 women). Among these stroke mortalities, 952 were hemorrhagic stroke (432 men and 520 women), and 957 were cerebral infarction (520 men and 437 women). The number of deaths from causes other than stroke was 18,868 (10,731 men and 8137 women); 5493 (2022 men and 3471 women) dropped out from the follow-up (5.8%); 67,349 (25,281 men and 42,068 women) were censored at the end of follow-up (71.4%). The medians (interquartile range, IQR) of follow-up years for hemorrhagic stroke mortality were 9.9 (5.4, 12.3) in men and 10.9 (5.9, 15.1) in women; the medians (IQR) of follow-up years for cerebral infarction mortality were 11.2 (7.1, 15.3) and 11.8 (7.9, 16.4) in men and women, respectively. Age-adjusted stroke mortality rates for each category of milk intake frequency was estimated to be 1.8, 2.0, 1.7, 1.6, an 1.5 per 1000 person-year for men and 1.3, 1.4, 1.2, 1.1, and 1.2 per 1000 person-year for women.

As listed in Table 1, compared with those who chose “never” as their milk intake frequency at baseline, milk drinkers were less likely to be a current smoker or a daily alcohol consumer in both men and women. Furthermore, people who consumed milk more than 1–2 times/month were more likely to be a daily consumers of vegetables and fruit, as well as coffee, and were more likely to perform exercise more than 1 h/week among both sexes.

Detailed results from the Bayesian survival models (crude, age-adjusted and multivariable-adjusted) according to the frequency of milk intake separated by sex are listed in Table 2 (men) and Table 3 (women). Compared to those who never had milk, both men and women had slower speed and lower hazard of dying from total stroke in crude models. Velocities that milk consumers dying from stroke were slowed down by a crude acceleration factor (AF) between 0.79 (SD = 0.05; 95% CrI: 0.74, 0.90) and 0.93 (SD = 0.04; 95% CrI: 0.85, 1.02) compared with non-consumers. Chances that the posterior crude HRs were estimated to be lower than 1 for those who had at least 1–2 times/month was higher than 86.5% in men and greater than 94.6% in women. However, lower hazard and delayed time-to-event was observed to remain after age or multivariable adjustment only among daily male milk consumers. Specifically, the means (SD; 95% CrI) of posterior multivariable-adjusted AF and HR for daily male consumers of milk were 0.88 (SD = 0.05; 95% CrI: 0.81, 0.96) and 0.80 (SD = 0.07; 95% CrI: 0.69, 0.93) with a probability of 99.0% to be smaller than the null value (=1). Daily female milk consumers had posterior AFs and HRs that were distributed with means of 0.97 (SD = 0.09; 95% CrI: 0.88, 1.10) and 0.95 (SD = 0.12; 95% CrI: 0.80, 1.17) which had about a 78.0% chance that their HRs could be smaller than 1.

Posterior distributions of AFs and HRs for mortality from hemorrhagic stroke were found to contain the null value for either men or women among all fitted models. In contrast, men who a had milk intake frequency higher than 1–2 times/week were found to be associated with an average of 17–20% slower velocity or 28–39% lower risk of dying from cerebral infarction compared to men who never drank milk (Model 2 in Table 2). Probability that the posterior HRs distributed below the null value was greater or equal to 97.5%. No evidence was found about the associations between milk intake and risk of cerebral infarction mortality among women.

## 4. Discussion

In the JACC study cohort, our analyses showed that men in Japan who consumed milk almost daily had lower hazard of dying from stroke especially from cerebral infarction. Our evidence also suggested that stroke mortality events were delayed among Japanese male daily milk consumers compared with non-consumers.

These findings showed similar negative effect estimates that were reported previously [16] using data from the same cohort, in which the outcomes of interest were focused on cardiovascular diseases and all-cause mortality. Moreover, we have further updated the results with more comprehensive and straightforward evidence about whether and how certain the data shown about the daily consumption of milk were contributing to a postponed stroke (mostly cerebral infarction) mortality event among Japanese men. A recent dose-response meta-analysis of 18 prospective cohort studies showed a similar negative association [2] between milk consumption and risk of stroke. The same meta-analysis also reported a greater reduction in risk of stroke (18%) for East Asian populations in contrast with the 7% less risk in the pooled overall finding for all populations combined. Benefits of increased milk intake might be particularly noticeable in East Asian countries where strokes are relatively more common, and milk consumption is much lower than those studies conducted among European or American populations [22]. However, the Life Span Study [6], which was conducted about 10 years earlier than the JACC Study, reported a null association between milk intake and fatal stroke among men and women combined who survived the Hiroshima and Nagasaki atomic bomb. The difference could be largely explained by the different targeting populations between the two studies. Kondo et al. [23] also reported a null association in both men or women from the NIPPON DATA80 study. In fact, the exact number of stroke mortality events was about 85% less and the number of participants in the NIPPON DATA80 study was about 90% less than those in the JACC study database—their null association might likely due to its limited statistical power. Umesawa et al. [24] reported that dairy calcium intake was inversely associated with ischemic stroke mortality risk, which could be considered as one of the potential pieces of evidence that supported our findings. Furthermore, a stronger inverse association between milk consumption and stroke mortality in men rather than in women was found in the Singapore Chinese Health Study [17] as well. The reasons for this gender difference is currently unknown. This might be due to generally better/healthier lifestyles (such as much less smokers and alcohol drinkers) in women regardless of milk intake frequency, or maybe due to other factors that were not available in/considered by our models, such as intake of calcium or other supplements. The probably existing beneficial effect of milk intake might be less evident in women than in men. Further investigation is needed.

Possible reasons for a protective effect of milk consumption against stroke could be interpreted, as such an association might be mediated by its content in calcium, magnesium, potassium, and other bioactive compounds, as recommended by Iacoviello et al. [25]. Apart from the inorganic minerals in milk that would be helpful with health effects, recent studies on animal models also indicated key evidence that stroke-associated morbidity was delayed in stroke-prone rats who were fed with milk-protein enriched diets [26,27]. More precisely, Singh et al. [28] found that whey protein and its components—lactalbumin and lactoferrin—improved energy balance and glycemic control against the onset of neurological deficits associated with stroke. Bioactive peptides from milk proteins were also responsible for the limitation of thrombosis [29] through their angiotensin convertase enzyme inhibitory potential, which might partly explain why the effect was found mainly for mortality from cerebral infarction in the current study.

### Strength and Limitations

Some limitations are worth mentioning here. First, the milk intake frequency, as well as other lifestyle information, was collected only once at the baseline and was self-reported. Apparently, life habits are possible to alter over time and these would result in misclassification and residual confounding. Second, despite the fact that the reasonable validity of FFQ in the JACC study cohort was assessed and confirmed, measurement errors are inevitable. Therefore, we did not try to compute the amount of consumption by multiplying an average volume per occasion with the frequency of intake, since the random error might be exaggerated and the observed associations may have attenuated. Third, the Bayesian way of conducting the survival analysis using a large sample size is computationally expensive. However, the classic maximum likelihood estimations are based on unrealistic assumptions, such as repeatability of the study, and fixed but unknown values of parameters, which are unfulfilled in a large cohort study in the current setting. The strengths of our analyses included that we transformed the research questions to more transparent ones that are easier for interpretation. Direct probabilities that daily milk intake is associated with lower hazard or delayed stroke mortality event were provided here after thorough computer simulations.

## 5. Conclusions

In conclusion, the JACC study database provided evidence that Japanese men who consumed milk daily had a lower risk of dying from stroke, particularly cerebral infarction, compared with their counterparts who never consumed milk. The time before an event of stroke mortality occurred was slowed down among men who drank milk regularly.

## Figures and Tables

**Table 1 nutrients-12-02743-t001:** Sex-specific baseline characteristics according to the frequency of milk intake (JACC study, 1988–2009).

			Milk Drinkers			
	Never	Drinkers	1–2 Times/	1–2 Times/	3–4 Times/	Almost
			Month	Week	Week	Daily
**Men (*n* = 39,386)**						
Number of subjects	8508	30,878	3522	5928	5563	15,865
Age, year (mean (SD))	56.8 (9.9)	56.8 (10.2)	55.2 (10.1)	55.4 (10.1)	55.4 (9.9)	58.1 (10.1)
Current smoker, %	58.7	49.8	57.4	55.9	51.1	45.4
Daily alcohol drinker, %	51.9	47.8	50.9	48.4	48.6	46.5
BMI, kg/m^2^ (mean (SD))	22.6 (3.4)	22.7 (3.4)	22.8 (2.8)	22.8 (2.8)	22.9 (5.4)	22.6 (2.8)
Exercise (>1 h/week), %	19.0	27.6	26.5	25.0	25.5	29.5
Sleep duration, 8–9 h, %	35.6	35.9	34.6	36.2	35.1	36.3
Energy intake, kcal/day (mean (SD))	1611 (505)	1764 (504)	1606 (496)	1679 (495)	1772 (509)	1830 (495)
Vegetable intake, daily, %	21.3	25.4	20.1	20.4	20.8	30.1
Fruit intake, daily, %	14.8	22.4	15.4	16.3	17.3	28.1
Green tea intake, daily, %	76.5	79.2	79.9	78.3	77.9	79.8
Coffee intake, daily, %	43.8	50.7	50.5	48.0	47.5	52.9
Educated over 18 years old, %	9.9	14.1	12.4	12.8	11.1	15.9
History of diabetes, %	5.0	6.3	4.5	4.2	5.5	7.7
History of hypertension, %	18.4	17.9	17.5	17.1	16.8	18.7
History of kidney diseases, %	3.0	3.4	3.8	3.0	3.0	3.5
History of liver diseases, %	5.8	6.5	6.3	6.0	5.4	7.2
**Women (*n* = 54,999)**						
number of subjects	10,407	44,592	3640	7590	8108	25,254
Age, year (mean (SD))	58.0 (10.2)	56.9 (9.9)	56.5 (10.2)	55.6 (10.1)	55.6 (9.9)	57.9 (9.9)
Current smoker, %	6.9	4.2	6.1	5.5	4.3	3.5
Daily alcohol drinker, %	4.3	4.5	5.5	4.3	4.2	4.6
BMI, kg/m^2^ (mean (SD))	23.0 (3.4)	22.9 (3.7)	23.0 (3.8)	23.1 (4.4)	23.1 (3.1)	22.8 (3.6)
Exercise (>1 h/week), %	13.6	20.8	17.1	18.5	18.8	22.6
Sleep duration, 8–9 h, %	27.7	25.6	25.1	25.9	25.4	25.7
Energy intake, kcal/day (mean (SD))	1519 (451)	1690 (449)	1522 (447)	1596 (443)	1661 (432)	1752 (443)
Vegetable intake, daily, %	24.7	30.4	25.0	24.6	24.2	34.8
Fruit intake, daily, %	25.0	35.7	26.6	29.2	29.2	41.1
Green tea intake, daily, %	73.8	76.8	77.0	76.4	75.8	77.3
Coffee intake, daily, %	39.6	48.2	46.2	46.4	44.4	50.2
Educated over 18 years old, %	4.8	8.3	6.7	7.2	6.5	9.4
History of diabetes, %	2.6	3.7	3.2	2.7	2.7	4.4
History of hypertension, %	21.5	19.7	20.5	19.1	18.9	20.0
History of kidney diseases, %	3.6	4.1	3.9	3.7	3.7	4.4
History of liver diseases, %	3.5	4.6	4.9	3.9	3.9	5.0

Note: Abbreviations: JACC: Japan Collaborative Cohort; SD: standard deviation; BMI: body mass index.

**Table 2 nutrients-12-02743-t002:** Summary of posterior Acceleration Factors (AF) and Hazard Ratios (HR) of mortality from total stroke, stroke types according to the frequency of milk intake in men (JACC study, 1988–2009).

	Never	1–2 Times/Month	1–2 Times/Week	3–4 Times/Week	Almost Daily
Person-year	135,704	56,551	97,098	92,153	252,364
N	8508	3522	5928	5563	15,865
Total Stroke	326	122	181	177	546
**Model 0**					
Mean AF (SD)	1	0.93 (0.07)	0.83 (0.05)	0.85 (0.05)	0.93 (0.04)
95% CrI	-	(0.81, 1.06)	(0.73, 0.94)	(0.74, 0.96)	(0.85, 1.02)
Mean HR (SD)	1	0.89 (0.09)	0.77 (0.07)	0.79 (0.07)	0.90 (0.06)
95% CrI	-	(0.73, 1.08)	(0.63, 0.91)	(0.66, 0.94)	(0.79, 1.03)
Pr(HR < 1)	-	86.5%	99.9%	99.7%	93.5%
**Model 1**					
Mean AF (SD)	1	0.99 (0.06)	0.90 (0.05)	0.91 (0.05)	0.85 (0.04)
95% CrI	-	(0.87, 1.11)	(0.81, 1.00)	(0.82, 1.01)	(0.78, 0.92)
Mean HR (SD)	1	0.98 (0.11)	0.84 (0.08)	0.86 (0.08)	0.76 (0.05)
95% CrI	-	(0.79, 1.19)	(0.70, 1.00)	(0.71, 1.02)	(0.66, 0.87)
Pr(HR < 1)	-	58.7%	97.3%	96.1%	100.0%
**Model 2**					
Mean AF (SD)	1	1.00 (0.07)	0.92 (0.06)	0.94 (0.06)	0.88 (0.05)
95% CrI	-	(0.88, 1.14)	(0.82, 1.03)	(0.84, 1.05)	(0.81, 0.96)
Mean HR (SD)	1	1.01 (0.12)	0.87 (0.09)	0.90 (0.09)	0.80 (0.07)
95% CrI	-	(0.81, 1.24)	(0.72, 1.05)	(0.74, 1.08)	(0.69, 0.93)
Pr(HR < 1)	-	50.6%	93.7%	89.6%	99.0%
Hemorrhagic stroke	100	42	58	56	176
**Model 0**					
Mean AF (SD)	1	1.03 (0.17)	0.85 (0.12)	0.87 (0.13)	0.98 (0.11)
95% CrI	-	(0.74, 1.38)	(0.63, 1.12)	(0.65, 1.14)	(0.78, 1.22)
Mean HR (SD)	1	1.03 (0.19)	0.82 (0.14)	0.84 (0.15)	0.97 (0.13)
95% CrI	-	(0.70, 1.46)	(0.56, 1.14)	(0.60, 1.17)	(0.75, 1.26)
Pr(HR < 1)	-	47.2%	88.4%	86.3%	63.1%
**Model 1**					
Mean AF (SD)	1	1.08 (0.17)	0.91 (0.13)	0.92 (0.13)	0.90 (0.10)
95% CrI	-	(0.80, 1.45)	(0.70, 1.20)	(0.71, 1.19)	(0.74, 1.11)
Mean HR (SD)	1	1.11 (0.21)	0.88 (0.16)	0.90 (0.16)	0.88 (0.12)
95% CrI	-	(0.75, 1.58)	(0.63, 1.25)	(0.63, 1.24)	(0.67, 1.14)
Pr(HR < 1)	-	31.6%	79.7%	76.6%	87.6%
**Model 2**					
Mean AF (SD)	1	1.11 (0.18)	0.93 (0.15)	0.96 (0.16)	0.96 (0.13)
95% CrI	-	(0.79, 1.58)	(0.70, 1.25)	(0.71, 1.34)	(0.76, 1.25)
Mean HR (SD)	1	1.14 (0.22)	0.92 (0.17)	0.95 (0.18)	0.95 (0.14)
95% CrI	-	(0.75, 1.61)	(0.63, 1.29)	(0.65, 1.37)	(0.71, 1.27)
Pr(HR < 1)	-	28.8%	72.4%	64.4%	69.3%
Cerebral infarction	151	41	64	66	198
**Model 0**					
Mean AF (SD)	1	0.76 (0.09)	0.71 (0.07)	0.74 (0.08)	0.79 (0.06)
95% CrI	-	(0.59, 0.94)	(0.58, 0.86)	(0.61, 0.89)	(0.68, 0.93)
Mean HR (SD)	1	0.65 (0.12)	0.59 (0.09)	0.64 (0.09)	0.71 (0.09)
95% CrI	-	(0.46, 0.92)	(0.43, 0.79)	(0.47, 0.85)	(0.56, 0.89)
Pr(HR < 1)	-	99.1%	99.9%	99.7%	99.5%
**Model 1**					
Mean AF (SD)	1	0.83 (0.08)	0.79 (0.07)	0.82 (0.07)	0.74 (0.05)
95% CrI	-	(0.68, 1.01)	(0.67, 0.93)	(0.69, 0.96)	(0.66, 0.84)
Mean HR (SD)	1	0.73 (0.13)	0.65 (0.10)	0.70 (0.11)	0.58 (0.07)
95% CrI	-	(0.49, 1.02)	(0.48, 0.88)	(0.51, 0.94)	(0.46, 0.72)
Pr(HR < 1)	-	96.9%	99.8%	98.9%	100.0%
**Model 2**					
Mean AF (SD)	1	0.84 (0.09)	0.80 (0.08)	0.83 (0.08)	0.75 (0.06)
95% CrI	-	(0.67, 1.02)	(0.67, 0.95)	(0.69, 0.99)	(0.66, 0.85)
Mean HR (SD)	1	0.73 (0.14)	0.67 (0.11)	0.72 (0.12)	0.61 (0.08)
95% CrI	-	(0.50, 1.04)	(0.48, 0.91)	(0.52, 0.99)	(0.48, 0.79)
Pr(HR < 1)	-	96.1%	99.1%	97.5%	99.8%

Note: Abbreviations: N, number of subjects,; SD, standard deviation; CrI, credible interval. Pr(HR < 1) indicates the probability for posterior HR to be smaller than 1. Model 0 = Crude model; Model 1 = age-adjusted model; Model 2 = multivariable adjusted model. Covariates included in Model 2: age, smoking habit, alcohol intake, body mass index, history of hypertension, diabetes, kidney/liver diseases, exercise, sleep duration, quartiles of total energy intake, coffee intake, and education level.

**Table 3 nutrients-12-02743-t003:** Summary of posterior Acceleration Factors (AF) and Hazard Ratios (HR) of mortality from total stroke and stroke types according to the frequency of milk intake in women (JACC study, 1988–2009).

	Never	1–2 Times/Month	1–2 Times/Week	3–4 Times/Week	Almost Daily
Person-year	173,222	59,904	129,233	139,919	418,925
N	10,407	3640	7590	8108	25,254
Total Stroke	300	84	182	172	585
**Model 0**					
Mean AF (SD)	1	0.88 (0.07)	0.87 (0.05)	0.79 (0.05)	0.88 (0.04)
95% CrI	-	(0.75, 1.03)	(0.78, 0.98)	(0.71, 0.90)	(0.80, 0.96)
Mean HR (SD)	1	0.83 (0.10)	0.81 (0.08)	0.70 (0.07)	0.81 (0.07)
95% CrI	-	(0.64, 1.05)	(0.68, 0.97)	(0.58, 0.85)	(0.71, 0.93)
Pr(HR < 1)	-	94.6%	98.7%	99.9%	99.6%
**Model 1**					
Mean AF (SD)	1	0.99 (0.09)	1.11 (0.08)	1.02 (0.08)	0.95 (0.06)
95% CrI	-	(0.85, 1.17)	(0.97, 1.26)	(0.89, 1.16)	(0.86, 1.06)
Mean HR (SD)	1	1.00 (0.14)	1.18 (0.14)	1.03 (0.12)	0.92 (0.09)
95% CrI	-	(0.76, 1.31)	(0.95, 1.47)	(0.82, 1.28)	(0.78, 1.09)
Pr(HR < 1)	-	52.3%	6.3%	42.0%	86.8%
**Model 2**					
Mean AF (SD)	1	1.01 (0.12)	1.11 (0.14)	1.02 (0.12)	0.97 (0.09)
95% CrI	-	(0.85, 1.20)	(0.97, 1.30)	(0.89, 1.19)	(0.88, 1.10)
Mean HR (SD)	1	1.01 (0.17)	1.19 (0.15)	1.03 (0.15)	0.95 (0.12)
95% CrI	-	(0.75, 1.36)	(0.96, 1.52)	(0.81, 1.31)	(0.80, 1.17)
Pr(HR < 1)	-	52.8%	6.4%	44.4%	78.0%
Hemorrhagic stroke	108	27	78	76	231
**Model 0**					
Mean AF (SD)	1	0.78 (0.13)	0.98 (0.12)	0.90 (0.11)	0.92 (0.09)
95% CrI	-	(0.55, 1.06)	(0.76, 1.25)	(0.70, 1.13)	(0.76, 1.12)
Mean HR (SD)	1	0.73 (0.16)	0.98 (0.15)	0.87 (0.14)	0.89 (0.11)
95% CrI	-	(0.47, 1.08)	(0.71, 1.31)	(0.64, 1.16)	(0.71, 1.15)
Pr(HR < 1)	-	94.7%	58.1%	83.1%	83.0%
**Model 1**					
Mean AF (SD)	1	0.88 (0.13)	1.12 (0.13)	1.04 (0.13)	0.95 (0.09)
95% CrI	-	(0.63, 1.17)	(0.90, 1.41)	(0.82, 1.32)	(0.80, 1.14)
Mean HR (SD)	1	0.84 (0.18)	1.17 (0.18)	1.06 (0.17)	0.93 (0.12)
95% CrI	-	(0.54, 1.24)	(0.86, 1.58)	(0.76, 1.45)	(0.73, 1.19)
Pr(HR < 1)	-	81.6%	16.9%	38.9%	74.6%
**Model 2**					
Mean AF (SD)	1	0.93 (0.24)	1.23 (0.38)	1.14 (0.33)	1.04 (0.25)
95% CrI	-	(0.64, 1.33)	(0.93, 1.98)	(0.87, 1.83)	(0.83, 1.55)
Mean HR (SD)	1	0.89 (0.22)	1.26 (0.26)	1.15 (0.23)	1.02 (0.19)
95% CrI	-	(0.55, 1.39)	(0.90, 1.90)	(0.83, 1.74)	(0.78, 1.51)
Pr(HR < 1)	-	73.2%	9.5%	24.8%	53.3%
Cerebral infarction	102	35	63	50	187
**Model 0**					
Mean AF (SD)	1	1.01 (0.13)	0.90 (0.09)	0.75 (0.08)	0.86 (0.06)
95% CrI	-	(0.79, 1.27)	(0.75, 1.10)	(0.60, 0.91)	(0.75, 0.99)
Mean HR (SD)	1	1.03 (0.20)	0.85 (0.14)	0.61 (0.11)	0.78 (0.10)
95% CrI	-	(0.69, 1.48)	(0.60, 1.13)	(0.43, 0.84)	(0.59, 0.99)
Pr(HR < 1)	-	51.9%	75.6%	97.6%	96.1%
**Model 1**					
Mean AF (SD)	1	1.21 (0.32)	1.16 (0.30)	0.98 (0.19)	0.97 (0.14)
95% CrI	-	(0.95, 2.08)	(0.93, 1.95)	(0.79, 1.48)	(0.84, 1.43)
Mean HR (SD)	1	1.37 (0.33)	1.25 (0.28)	0.94 (0.22)	0.92 (0.17)
95% CrI	-	(0.89, 2.18)	(0.87, 1.95)	(0.63, 1.52)	(0.69, 1.40)
Pr(HR < 1)	-	8.5%	14.2%	70.1%	79.4%
**Model 2**					
Mean AF (SD)	1	1.19 (0.19)	1.12 (0.15)	0.96 (0.12)	0.97 (0.09)
95% CrI	-	(0.94, 1.62)	(0.92, 1.49)	(0.78, 1.21)	(0.83, 1.18)
Mean HR (SD)	1	1.38 (0.29)	1.21 (0.22)	0.91 (0.18)	0.94 (0.14)
95% CrI	-	(0.89, 2.02)	(0.85, 1.70)	(0.62, 1.34)	(0.69, 1.25)
Pr(HR < 1)	-	7.3%	15.6%	72.8%	70.0%

Note: Abbreviations: N, number of subjects; SD, standard deviation; CrI, credible interval. Pr(HR < 1) indicates the probability for posterior HR to be smaller than 1. Model 0 = Crude model; Model 1 = age-adjusted model; Model 2 = multivariable adjusted model. Covariates included in Model 2: age, smoking habit, alcohol intake, body mass index, history of hypertension, diabetes, kidney/liver diseases, exercise, sleep duration, quartiles of total energy intake, coffee intake, and education level.

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
