# Peer review of "Milk Intake and Stroke Mortality in the Japan Collaborative Cohort Study—A Bayesian Survival Analysis"

_nutrients, 2020, doi:10.3390/nu12092743_

Round 1
Reviewer 1 Report
Review of article
Milk intake and stroke mortality in the Japan Collaborative Cohort Study
- a Bayesian survival analysis
Thank you very much to the Editorial for entrusting me with the task of reviewing the article. The manuscript presented for review raises a very important health problem, which is stroke. Stroke continues to be one of the leading causes of death and disability in adults. It is
a significant health problem which contributes significantly to reduce the quality of life. The key issue from the medical and social point of view remains effective primary and secondary prevention. The current state of knowledge indicates a significant and constantly growing importance non-pharmacological activities undertaken as part of primary prevention stroke, representing a complementary for drug therapy and just as important area of intervention. The broadly understood "primary prevention” has to be implementing it at an early stage, raising awareness of healthcare professionals and patients
It is known that what you eat can increase or decrease your risk of stroke. The relationship between milk intake and stroke mortality is very interesting and it has a practical effect.The authors of the presented manuscript, as well as other researchers, observed that higher milk consumption is also associated with a healthier diet, greater physical activity and not smoking.
The summary actually shows the problem at hand.
The work structure is correct. Most sections are clearly and accurately described.
Tables are legible and correctly described.
The writings cited by the authors are up-to-date and appropriate.
Minor revision:
I would make a clearer and separate section for "Conclusions" and "Study limitations".
Congratulations to the authors of the interestingly described research on a large group of participants.
Reviewer 2 Report
The manuscript by C Wang et al includes a cohort of 94385 men and women from Japan, free from self-reported history of stroke, cancer, myocardial infarction and other types of heart disease at baseline who answered an FFQ regarding milk consumption. Between baseline in 1988-90 and end of follow-up in 2009, 2675 deaths from stroke were recorded on death registers. The authors applied Bayesian survival analysis to calculate hazard ratios and acceleration factors. With a range of milk consumption from “never” to “almost daily”, a higher milk consumption was associated with lower hazard and speed from dying from total stroke among men but not among women.
I have the following comments and questions for the authors:
- The introduction and discussion should however be more clear about the fact that the study is being performed in the lower range of the milk intake distribution. The dose-response meta-analysis in reference 2 indicates that the association may be non-linear. It would be good if the authors in more detail compared their results with other studies using fatal stroke as outcome and perhaps also focus on those cohorts with a low milk intake. For example, please elaborate on references 6, 20-23 in the discussion. Some of the references do not investigate milk intake per se. In the discussion, please specify that reference 19 does not investigate stroke mortality but CVD and all-cause mortality.
- I am unfortunately not familiar with the Bayesian models used in this work. I am sure that many readers will be in a similar situation as me. I appreciate that these models can calculate hazard ratios without fulfilling the proportional hazards assumption. The authors should in more detail describe the benefits and limitations of these models in relation to the more commonly used Cox proportional hazards regression. Is a HR estimated from these models directly comparable with that from a Cox model? How is an acceleration factor interpreted? Do the models assume that a higher milk intake is associated with a lower stroke mortality, indicated also by the Pr(HR<1)? If so, how would that influence the validity of the method since you in the introduction write that there are reports of inverse associations, no association, and direct associations? In Table 2, both SD and 95% CrI are given. Is there a reason for specifying the SD?
- Why were analyses performed stratified by sex? Since no pooled analysis is presented, the reason for a stratified analysis should be stated in the aims, unless this was a secondary analysis and then also the results from the total cohort should be presented. The authors should also discuss the observed sex differences in more detail. Do other studies show similar sex differences? What potential biological mechanisms may explain them?
- The aim states to “provide a more straightforward answer to the primary research question” – compared to what? And how is this more straightforward answer achieved. This is not clear, neither in the aim nor in the discussion (lines166-167). Please revise.
- Please describe the context of milk intake in Japan. Is milk consumed as a beverage with food, added to coffee or other consumption patterns? Are other milk products than fresh non-fermented milk consumed and could be included in the “milk” exposure?
- Please specify age at baseline in abstract and methods.
- Outcome ascertainment: Information on cause of death was from death certificates. Is there a possibility of individual linkage of the FFQ responses to a cause of death registry or were each of the participants’ death certificates reviewed? How many died from other causes, how many moved from the study area (and were thus lost to follow-up) and how many were censored at the end of follow-up (alive). Has stroke mortality changed during follow-up? Is there a difference in time trends of stroke subtype mortality? In how many cases was stroke subtype not recorded? The authors should also discuss potential reasons for differences in the results for stroke subtypes.
- I agree with the authors that presenting the results in the FFQ categories is a good option, rather than to calculate the amount of consumption. However, I wonder if there is some indication from the validation study as to how much is on average consumed in the higher categories of intake?
- Why did you adjust for sleep duration, is that a factor that influences both milk intake and stroke mortality risk?
- The manuscript has minor language errors throughout and would benefit from a language review.
Round 2
Reviewer 2 Report
I would like to thank the authors for their responses and amendments to the manuscript. I have no further comments.